# Solvent-Free Synthesis of MgO-Modified Biochars for Phosphorus Removal from Wastewater

**DOI:** 10.3390/ijerph19137770

**Published:** 2022-06-24

**Authors:** Siyu Xu, De Li, Haixin Guo, Haodong Lu, Mo Qiu, Jirui Yang, Feng Shen

**Affiliations:** 1Agro-Environmental Protection Institute, Ministry of Agriculture and Rural Affairs, No. 31 Fukang Road, Nankai District, Tianjin 300191, China; 82101202147@caas.cn (S.X.); lide@webmail.hzau.edu.cn (D.L.); haixin.guo.c8@tohoku.ac.jp (H.G.); qiumo@caas.cn (M.Q.); 2College of Resources and Environment, Huazhong Agricultural University, No. 1, Shizishan Street, Hongshan District, Wuhan 430070, China; 3Department of Chemical Engineering, University of Alberta, Edmonton, AB T6G 1H9, Canada; haodong1@ualberta.ca

**Keywords:** phosphorus adsorption, ball milling, biochar modification, magnesium oxide

## Abstract

Adsorption is an efficient technology for removing phosphorus from wastewater to control eutrophication. In this work, MgO-modified biochars were synthesized by a solvent-free ball milling method and used to remove phosphorus. The MgO-modified biochars had specific surface areas 20.50–212.65 m^2^ g^−1^ and pore volume 0.024–0.567 cm^3^ g^−1^. The as-prepared 2MgO/BC-450-0.5 had phosphorus adsorption capacities of 171.54 mg g^−1^ at 25 °C and could remove 100% of phosphorus from livestock wastewater containing 39.51 mg L^−^^1^ phosphorus. The kinetic and isotherms studied show that the pseudo-second-order model (*R*^2^ = 0.999) and Langmuir models (*R*^2^ = 0.982) could describe the adoption process well. The thermodynamic analysis indicated that the adsorption of phosphorus on the MgO-modified biochars adsorbent was spontaneous and endothermic. The effect of pH, FTIR spectra and XPS spectra studies indicated that the phosphorus adsorption includes a protonation process, electrostatic attraction and precipitation process. This study provides a new strategy for biochar modification via a facile mechanochemical method.

## 1. Introduction

Phosphorus is an indispensable nutrient element in the growth process of aquatic plants [1,2], existing in various forms depending on the pH values of water, such as PO_4_^3−^, HPO_4_^2−^ and H_2_PO_4_^−^. However, excessive discharge of phosphorus from agricultural, industrial and domestic sources into water will cause eutrophication, damage the ecological environment and even threaten human health. According to the Integrated Wastewater Discharge Standard (GB 8978-1996) established in China, phosphate discharge (in terms of P) is divided into the primary standard (≤0.5 mg L^−1^) and the secondary standard (≤1.0 mg L^−1^). In addition, the US Environmental Protection Agency (EPA) recommends that the total phosphorus concentration in surface water should not exceed 0.1 mg L^−1^, which is considered safe for aquatic life [3]. Therefore, excessive phosphorus needs to be removed from the water in time to meet the above standards.

At present, the methods for phosphorus removal mainly include biological methods, chemical methods and physical methods. Among them, biological methods such as assimilation [4] are often hindered by the lack of carbon sources and strict reaction conditions [5,6]. Chemical methods such as precipitation [7] are the most stable and efficient methods for phosphorus removal, but their high cost and toxicity limit their further application [8]. In contrast, physical methods such as the commonly used adsorption technique are much more economical and highly selective for removing phosphate from water, with less potential for secondary pollution. The adsorbents include zeolite [9], hydrotalcite [10], Al_2_O_3_ [11], biochar [12] and so on. Among them, biochar is considered to be a potential adsorbent for water environmental governance due to its low cost and environmental friendliness [13].

Generally, pristine biochar prepared by direct pyrolysis of biomass has poor adsorption performance for phosphorus due to its negatively charged surface, the low specific surface area and limited adsorption sites [14,15]. Therefore, it is of great significance to develop biochar modification approaches for improving its adsorption capacity. Loading metal oxides such as MgO [16], CaO [17], La_2_O_3_ [18] on the surface of biochar using the impregnation method is the most common modification process, in light of the great adsorption performance of metal oxides on the removal of phosphate. Many studies have shown that MgO demonstrates good adsorption performance on the removal of PO_4_^3−^ from water [19,20,21]. For example, Li et al. [22] used the impregnation method for the preparation of MgO-modified biochar with the maximum phosphorus adsorption capacity of 121.25 mg g^−1^. Compared with traditional impregnation, which involves introducing metal oxides onto the surface of biochar, the emerging ball milling method is a more facile strategy due to its solvent-free modification approach [23,24].

Ball milling has emerged as a favorable method for the synthesis of nanomaterials and is becoming recognized as an environmentally friendly and low-cost method for enhancing the physicochemical properties of metal oxides/biochar nanomaterials [25,26,27]. The composites can be ground into nanoscale particles by ball milling, thereby increasing the specific surface area of the materials, which, in turn, enhances their adsorption capacity [28]. Many metal oxides/biochar composites, such as Fe_3_O_4_^−^ or CuO-modified biochar, have been synthesized using the ball milling method for the removal of tetracycline [29], methylene blue [30] and reactive red [31]. The results of these studies indicate that ball milling can increase the surface area and functional groups of biochars [32,33], thus endowing the metal oxides/biochar composites higher removal efficiency than that of pristine biochar. Inspired by the above studies, MgO-biochar composites were synthesized by the ball milling method followed by a pyrolysis process in this study and employed for phosphorus removal from wastewater. The effects of some key factors on phosphorus adsorption were evaluated, including the content of Mg precursor, pyrolysis temperature, ball milling time, solution pH, adsorbent dosage, co-existing ions and reaction temperature. A mechanism for phosphorus removal was also proposed.

## 2. Materials and Methods

### 2.1. Materials

Rice straw was obtained from Xiangtan in Hunan province (China). Potassium phosphate monobasic (KH_2_PO_4_, AR, 99.5%), potassium nitrate (KNO_3_, AR, 99.0%), potassium bicarbonate (KHCO_3_, AR, 99.5%) and potassium sulfate (K_2_SO_4_, AR, 99%) were purchased from Shanghai Macklin Biochemical Co., Ltd. (Shanghai, China). Potassium chloride (KCl, AR, ≥99.5%) were supplied by Tianjin JiangTian Chemical Technology Co., Ltd. (Tianjin, China). Magnesium acetate tetrahydrate (Mg(CH_3_COO)_2_·4H_2_O, AR, 99%) was provided by Yuanye Biological Technology Co., Ltd (Shanghai, China). The livestock wastewater used in this study was taken from a cattle farm in the Dali Experimental Station. Before use, the wastewater was pretreated to remove suspended solids through a 0.2 μm filter membrane. Ultrapure water was produced by the Direct-Q3 UV system.

### 2.2. Adsorbents Preparation

Rice straw was washed, dried, ground and finally sieved through a 60 meshes sieve (0.3 mm). In a typical run, Mg(CH_3_COO)_2_·4H_2_O (Mg precursor) and rice straw powder (Mg precursor: rice straw = 0.5:1, 1:1, 2:1 or 3:1 in mass ratio) were loaded into a ball milling tank with 27.3 g of ZrO_2_ balls (diameter 6.0 mm). Then, the tank was placed in a planetary ball mill (PM-100, RETSCH, Haan, Germany). The rotation speed of the planetary disk was set as 350 rpm. After ball milling for a certain period of time (0.25–10 h), the resulting ball mixing sample was placed in a tube furnace (OTF-1200X, Hefei Kejing Material Technology Co., Ltd., Hefei, China) and heated to the designed temperature (250–850 °C) with a heating rate of 2 °C min^−1^ under N_2_ atmosphere and maintained for 2 h. After pyrolysis treatment, the resulting biochar was ground to powder and denoted as xMgO/BC-y-z (where x, y and z represented Mg precursor-to-rice straw mass ratio, pyrolysis temperature and ball milling time, respectively). The biochar named BC was prepared by direct pyrolysis of pure rice straw at 450 °C after ball milling for 0.5 h.

### 2.3. Adsorbent Characterization

The morphology structure of the adsorbents was investigated by scanning electron microscopy (SEM). Surface element analysis was also conducted simultaneously with SEM at the same surface locations, using energy-dispersive X-ray spectroscopy (EDS, Octane Elect Super, EDAX, Mahwah, NJ, USA). X-ray diffraction (XRD, Ultima IV, Rigaku, Tokyo, Japan) was employed to determine the crystallographic structure of adsorbents in the range of 5° to 90°, operating at 40 kV and 40 mA. X-ray photoelectron spectroscopy (XPS, Escalab 250xi, Thermo Fisher, Waltham, MA, USA) was used to determine the elemental composition and chemical state of adsorbent surfaces with mono Al Kα radiation. N_2_ adsorption–desorption isotherms were obtained from ASAP 2460 instrument (Micromeritics, Atlanta, GA, USA). The specific surface area (S_BET_) and pore size (D_BJH_) of adsorbents were calculated by the BET method and the BJH method. The total pore volume (V_total_) was determined from the amount of adsorbed N_2_ at a relative pressure (P/P_0_) of 0.99. The Mg loadings of the adsorbents were determined with inductively coupled plasma optical emission spectroscopy (ICP-OES, ICP OES 730, Agilent, Tokyo, Japan), while the contents of C, H, O and N were measured by an elemental analyzer (UNICUBE, Elementar, Hanau, Germany). Thermo-gravimetric analysis (TGA) was performed by DSC/DTA-TG analyzer (STA 449 F5, NETZSCH, Bavaria, Germany) in N_2_ stream with a heating rate of 10 °C min^−1^ from 25 °C to 900 °C. The zeta potential was measured in solutions of different pH using a zeta potential analyzer (ZS90, Malvern Panalytical, Worcs, UK). Fourier transformed infrared spectroscopy (FTIR, Thermo Nicolet iS5, Thermo Fisher, Waltham, MA, USA) was performed to determine the functional group changes.

### 2.4. Adsorption Experiments

#### 2.4.1. Optimization of Material Preparation Conditions

In order to select the adsorbent with the best adsorption performance, a series of preparation conditions were screened, including Mg precursor-to-rice straw mass ratio (0, 0.5, 1, 2, 3), pyrolysis temperature (250, 450, 650, 850 °C) and ball milling time (0.25, 0.5, 2, 10 h). The adsorption experiment was conducted in the 50 mL polyethylene tube loaded with 20 mg of adsorbents and 40 mL of phosphorus solution (100 mg L^−1^, pH = 5.2) in a shaker at 250 rpm for 24 h. All adsorption experiments were run in triplicate, and the average values were reported with a relative deviation below 5%. The adsorption capacity *q*_e_ (mg g^−1^) of phosphorus on the adsorbent was calculated at equilibrium by the following Equation (1):(1)$$q_{e}=\frac{(C_{0}\;-\;C_{e})\;\times \;V}{m}$$
where *C*_0_ and *C*_e_ (mg L^−1^) are the initial and equilibrium concentrations of phosphorus in the solution, respectively; *V* (L) is the volume of phosphorus solution; *m* (g) is the weight of the adsorbent.

#### 2.4.2. The Adsorption Behavior of Phosphorus on MgO-Modified Biochars

Adsorption kinetics for phosphorus on different adsorbents were determined at different time intervals (2, 6, 8, 10, 12, 16, 18, 24, 36, 48, 60 h) by adding 20 mg of adsorbents to 40 mL of 100 mg L^−1^ phosphorus solution at 25 °C. Adsorption isotherms for phosphorus on the adsorbents were determined with different concentrations ranging from 0 to 200 mg L^−1^ (pH 4.6–6.8) at 25 °C for 24 h. Adsorption thermodynamics for phosphorus on 2MgO/BC-450-0.5 were conducted with different reaction temperatures (298–318 K) by adding 20 mg of adsorbents to 40 mL of 100 mg L^−1^ phosphorus solution for 24 h. The details of this process are presented in the Appendix A.

A series of adsorption experiments were carried out to evaluate the effects of environmental parameters on the performance of MgO-modified biochars for phosphorus removal. The effects of solution pH (1–11) on phosphorus removal were investigated by adding 20 mg of adsorbents to 40 mL of 100 mg L^−1^ phosphorus solution and adsorbing for 24 h at 25 °C. The effect of adsorbent dosage (0.25–1.25 g L^−1^) was studied with the initial phosphorus concentrations of 200 mg L^−1^ at pH = 5 and the rest of the conditions were kept constant. The effect of co-existing ions (0.01 M) was evaluated, including Cl^−^, NO_3_^−^, SO_4_^2−^ and HCO_3_^−^. The removal rate *R* (%) of phosphorus was calculated following Equation (2):(2)$$R=\frac{C_{0}\;-\;C_{e}}{C_{0}}\;\times \;100\%$$

### 2.5. Cyclic Performance Test

The reusability of 2MgO/BC-450-0.5 was tested by 5 adsorption–desorption cycle experiments. The adsorbent (80 mg) was treated with 40 mL phosphorus solution (10 mg P L^−1^) for 12 h at 250 rpm and 25 °C, and the removal rate (*R*) was calculated as above. The phosphorus-adsorbed sample was treated with 1 mol L^−1^ NaOH solution for 8 h for the desorption of phosphorus from 2MgO/BC-450-0.5. The regenerated 2MgO/BC-450-0.5 was again used for phosphorus adsorption under the same conditions.

### 2.6. Phosphorus Removal from Livestock Wastewater

To estimate the potential for treating livestock wastewater, the adsorption experiment was performed by mixing 2MgO/BC-450-0.5 (10–30 mg) with 40 mL of wastewater and shaking for 12 h as described above. After the mixtures were filtered, the phosphorus removal rate was calculated as above.

## 3. Results and Discussion

### 3.1. Characteristics of MgO-Modified Biochars

The specific surface area and porosity of biochars were determined by N_2_ adsorption–desorption experiment analysis. The isotherms results show that all MgO-modified biochars exhibited typical type-IV curves, which is the characteristic of dominated mesopores (Appendix A) [34]. This was possible due to the intercalation of MgO particles into the carbon skeleton, which might be unfavorable for the formation of micropores [35]. As Table 1 shows, the biochar without MgO modification (BC) had low S_BET_ (12.31 m^2^ g^−1^) and V_total_ (0.017 cm^3^ g^−1^). The surface area (S_BET_) and V_total_ were increased for MgO-modified biochars (Table 1). This is because the magnesium salts could be used as carbon activators to produce more pores and increase a specific surface area [22]. The 2MgO/BC-450-0.5 had the highest S_BET_ and V_total_, reaching up to 144.87 m^2^ g^−1^ and 0.197 cm^3^ g^−1^, respectively. However, both surface area (S_BET_) and V_total_ were slightly reduced for 3MgO/BC-450-0.5, which was prepared with a higher Mg precursor-to-rice straw mass ratio. The pores in 3MgO/BC-450-0.5 were blocked by the inserted MgO particles, leading to a decrease in S_BET_ compared with that of 2MgO/BC-450-0.5 (Table 1) [36,37]. The average dimeter (D_BJH_) decreased after mixing ball-milling of biochar and Mg precursor (Table 1). As can be seen from Appendix A, the surfaces of BC, 0.5MgO/BC-450-0.5 and 1MgO/BC-450-0.5 were relatively smooth, while the coarse surfaces were found in 2MgO/BC-450-0.5 and 3MgO/BC-450-0.5. Particularly, as Mg precursor-to-rice straw mass ratio was increased to 3:1, deposition of MgO particles was observed on the surface of the 3MgO/BC-450-0.5. EDS spectrum pattern of 2MgO/BC-450-0.5 confirmed that Mg and O species were distributed evenly on C species without aggregation (Appendix A).

### 3.2. Screening of MgO-Modified Biochars

#### 3.2.1. Effect of MgO Content

The adsorption performance of phosphorus with different adsorbents is given in Figure 1a. The phosphorus adsorption performance followed the order: 2MgO/BC-450-0.5 > 3MgO/BC-450-0.5 > 1MgO/BC-450-0.5 > 0.5MgO/BC-450-0.5 > BC (Figure 1a). In general, the adsorption ability of pristine biochar for phosphorus was relatively weak due to the negatively charged surfaces and low specific surface area [38,39]. After loading of MgO, the MgO-modified biochars showed much higher phosphorus adsorption capacity. However, adding excessive Mg precursor weakened the phosphorus adsorption performance of 3MgO/BC-450-0.5 due to the aggregation of MgO, which was in accordance with SEM analysis, thus resulting in a reduction in adsorption sites.

#### 3.2.2. Effect of Pyrolysis Temperature

TGA analysis was conducted on the ball-milled mixture of Mg precursor and straw with Mg precursor-to-rice straw mass ratio of 2. As shown in Figure 1b, the weight loss process could be roughly divided into three stages [40,41]: the first weight loss (75–170 °C) was due to removal of adsorbed molecular water. The second weight loss (170–380 °C) corresponded to decomposition of Mg(CH_3_COO)_2_·4H_2_O and hemicellulose. At the third stage (380–460 °C), the weight loss of only 5.09% was ascribed to the formation of MgO and degradation of cellulose and lignin. Then, four MgO-modified biochars were prepared at pyrolysis temperatures of 250, 450, 650 and 850 °C, respectively. The result of adsorption experiment showed that phosphorus adsorption capacity was rapidly increased from 17.08 to 168.85 mg g^−1^ when pyrolysis temperature increased from 250 to 450 °C (Figure 1c). However, further increase in the pyrolysis temperature to 850 °C caused a slight reduction in adsorption capacity. The S_BET_ of MgO-modified biochars was increased from 40.10 to 212.65 m^2^ g^−1^ from 250 to 850 °C (Table 1). The Mg elemental content in 2MgO/BC-250-0.5, 2MgO/BC-450-0.5, 2MgO/BC-650-0.5 and 2MgO/BC-850-0.5 was determined to be 12.72%, 27.62%, 28.89% and 31.16%, respectively (Appendix A). However, a higher pyrolysis temperature resulted in the agglomeration of MgO particles (Appendix A), thereby leading to a decrease in adsorption capacity. This result was consistent with the previous study [42].

#### 3.2.3. Effect of Ball Milling Time

Considering that ball milling time may affect the morphology and structure of the adsorbent [43,44], four ball milling times (0.25, 0.5, 2 and 10 h) were selected. The result of the adsorption experiment (Figure 1d) showed that 2MgO/BC-450-0.5 had the maximum phosphorus adsorption capacity (176.42 mg g^−1^). The morphologies (Appendix A) of MgO-modified biochars demonstrated that ball milling treatment for a short time could break the samples into small particles, but milling for a longer time could cause the samples to agglomerate together, due to the compaction of ball milling [45]. This was the probable reason why 2MgO/BC-450-2 and 2MgO/BC-450-10 had lower specific surface areas and smaller pore volumes (Table 1). Ball milling for 0.5 h was sufficient to endow the final MgO-modified biochar with the largest pore volume and highest specific surface area. In the pre-experiment, the mixture of Mg precursor and straw was directly pyrolyzed without ball milling. However, the phosphorus adsorption capacity of the prepared sample was very low (37.65 mg g^−1^), suggesting that the ball milling treatment is essential for improving the adsorption performance of the adsorbents. Based on the above studies, 2MgO/BC-450-0.5 prepared at the optimum conditions had the best phosphorus adsorption; they were then employed to study the adsorption behavior in detail.

### 3.3. Adsorption Kinetics, Isotherms and Thermodynamics

The adsorption kinetics experiments of BC and 2MgO/BC-450-0.5 were studied for a maximum of 60 h (Figure 2a). The adsorption of phosphorus on BC was much slower and mainly controlled by physical adsorption. The 2MgO/BC-450-0.5 provided fast adsorption for phosphorus with equilibrium < 24 h, followed by a slow adsorption process. The fast adsorption at the initial time could be attributed to the electrostatic attraction between the positively charged MgOH^+^ surface and negatively charged phosphate ions [46]. The slow adsorption was related to diffusion controlled adsorption and physical adsorption [47]. To understand how 2MgO/BC-450-0.5 worked for phosphorus adsorption, the kinetic data were fitted by the pseudo-first-order model and the pseudo-second-order model. The experimental data were fitted well by the pseudo-second-order kinetic model (*R*^2^ = 0.999) and pseudo-first-order kinetic model (*R*^2^ = 0.995). It was indicated that the adsorption behavior of phosphorus on 2MgO/BC-450-0.5 was mainly chemisorption predominated.

The adsorption isotherms of biochars were fitted with Freundlich and Langmuir models (Figure 2b). Compared with the Freundlich model (*R*^2^ = 0.762), the Langmuir model (*R*^2^ = 0.982) fitted the adsorption data better, suggesting that phosphorus adsorption on the surface of 2MgO/BC-450-0.5 was a monolayer adsorption process [48]. The obtained *q*_max_ on 2MgO/BC-450-0.5 based on the Langmuir model were 171.54 mg g^−1^, which was almost 27-times that of BC. The *q*_max_ of MgO-biochars for phosphorus adsorption surpassed most of MgO-modified materials and are comparable with some CaO-modified materials (Table 2). On the basis of MgO content (46.03%) and phosphorus adsorption capacity (*q*_max_ = 171.54 mg g^−1^), the mole ratio of MgO to phosphorus was 2:1, considering that only the surficial atoms of MgO particles were available for the adsorbate.

The adsorption thermodynamics of phosphorus on 2MgO/BC-450-0.5 were conducted as shown in Figure 2c. With the temperature increasing from 298 to 318 K, the *q*_max_ was increased from 171.54 to 212.19 mg g^−1^. According to the literature [49], the values of ∆*G*^0^, ∆*S*^0^ and ∆*H*^0^ were calculated. It could be observed from Figure 2d and Appendix A that the values of ∆*G*^0^ were negative and declined with the rise in temperature, revealing that phosphorus adsorption on 2MgO/BC-450-0.5 was spontaneous. The value of ∆*S*^0^ was positive, illustrating the high randomness during the adsorption process. Furthermore, the value of ∆*H*^0^ was positive, indicating an endothermic adsorption process. Consequently, a higher reaction temperature was conducive to phosphorus adsorption on 2MgO/BC-450-0.5.

### 3.4. Effect of Initial pH

pH is an important factor that affects both the forms of phosphate in the solution and the surface charge properties of the adsorbent. As displayed in Figure 3a, the phosphorus adsorption capacity of 2MgO/BC-450-0.5 drastically increased from 20.57 to 136.63 mg g^−1^ with initial solution pH varying from 1 to 3. Then, the adsorption capacity reached the maximum capacity (160.70 mg g^−1^) at pH 5. However, with the initial pH value rising from 5 to 11, the adsorption capacity decreased from 160.70 to 53.37 mg g^−1^. A rise in the final pH value was observed due to the release of OH^-^ from the reaction of MgO and water (MgO + H_2_O→MgOH^+^ + OH^−^) [54]. According to the pKa values of phosphate, phosphate existed in the forms of H_3_PO_4_, H_2_PO_4_^−^ and HPO_4_^2−^ at pH ≤ 2.12, 2.12–7.21 and 7.21–12.67, respectively [55,56]. The pH_PZC_ value of 2MgO/BC-450-0.5 was determined to be 5.19 (Figure 3b), suggesting that the surface of 2MgO/BC-450-0.5 was positively charged at a pH value of <5.19. Compared with H_3_PO_4_, the negative H_2_PO_4_^−^ was more easily adsorbed on 2MgO/BC-450-0.5 through electrostatic interaction and reacted with protonated MgO afterwards. By contrast, the surface of 2MgO/BC-450-0.5 was negatively charged at a pH value of >5.19, indicating that 2MgO/BC-450-0.5 repulsed H_2_PO_4_^-^ and HPO_4_^2−^. Moreover, OH^−^ competed with phosphate for the adsorption sites on 2MgO/BC-450-0.5. These reasons can lead to the decline in phosphate adsorption capacity at a higher initial pH.

### 3.5. Effect of Adsorbent Dosage

It was seen that phosphorus removal efficiency was increased from 16.74% to 93.46%, along with the increase in 2MgO/BC-450-0.5 dosage from 0.25 to 1.25 g L^−1^ (Figure 4). The maximum phosphorus adsorption capacity of 202.10 mg g^−1^ was obtained with the dosage of 0.75 g L^−1^. Generally, more adsorbents gave more sites for phosphorus adsorption. However, further raising of adsorbent content resulted in a decrease in the adsorption capacity per unit mass [57].

### 3.6. Effect of Co-Existing Ions

There are various inorganic ions in wastewater such as Cl^−^, NO_3_^−^, SO_4_^2−^ and HCO_3_^−^ that can interfere with phosphorus adsorption on 2MgO/BC-450-0.5 by competing for the adsorption sites. The effect of co-existing ions on phosphorus adsorption is given in Figure 5. Compared to the blank (without co-existing ions), the presence of Cl^−^, NO_3_^−^ and SO_4_^2−^ in the solution had almost no effect on phosphorus removal, whereas the existence of HCO_3_^−^ significantly reduced the phosphorus removal efficiency by 30.79%. The reason was that HCO_3_^−^ increased the solution pH, causing a decrease in the content of positive charge on the surface of the 2MgO/BC-450-0.5. On the other hand, the properties of HCO_3_^−^ were similar to those of phosphate, leading HCO_3_^−^ to compete with phosphate for active adsorption sites on the surface of 2MgO/BC-450-0.5 [58].

### 3.7. Adsorption Mechanism

To obtain more convincing evidence on the adsorption mechanism, FTIR spectra, XPS spectra and XRD patterns of 2MgO/BC-450-0.5 before and after phosphorus adsorption were conducted. As depicted in Figure 6a, changes in functional groups on the surface of 2MgO/BC-450-0.5 were induced by phosphorus adsorption. The broad bond at 3390 cm^−1^ corresponded to the stretching vibration of −OH [59], but the content of −OH increased markedly after adsorption, indicating that some products containing crystal water appeared on the adsorbent [60]. In FTIR spectra, a band at around 3700 cm^−1^ was assigned to free −OH on the surface of MgO, which disappeared after adsorption. This change can be a hint for the involution of MgO in phosphate adsorption. The peak at around 1060 cm^−1^ was assigned to the P–O asymmetry vibration [61]. After phosphorus adsorption reaction, the bond intensity became stronger, implying the adsorption of phosphorus on 2MgO/BC-450-0.5, which was consistent with the XRD result (Appendix A). The XRD patterns of 2MgO/BC-450-0.5 after adsorption showed that the characteristic peaks of MgO disappeared. In addition, the peaks of Mg–O stretching vibration at 500–750 cm^−1^ became weaker and a new peak attributed to the P–O bond appeared at 575 cm^−1^, proving that the phosphorus was adsorbed on the MgO surface [62].

XPS was carried out to analyze the composition changes in surface materials of 2MgO/BC-450-0.5. In Figure 6b, the full-scan spectra of 2MgO/BC-450-0.5 revealed that 2MgO/BC-450-0.5 was mainly composed of Mg, O and C, which was previously confirmed by EDS spectrum pattern (Appendix A). After adsorption, a new peak appeared at 135 eV corresponding to P 2p [63]. In order to clarify the form of adsorbed phosphorus on the surface of 2MgO/BC-450-0.5, XPS spectra of Mg 1s was performed before and after phosphorus adsorption. Before adsorption of phosphorus (Figure 6c), the spectrum was be divided into two peaks. The peaks at around 1305.4 and 1304.2 eV were attributed to MgO and Mg(OH)_2_, respectively [64]. After adsorption (Figure 6d), three peaks could be detected at 1306.8, 1305.0 and 1304.0 eV, which were ascribed to Mg(H_2_PO_4_)_2_, MgHPO_4_ and MgO, respectively [64,65].

The process of phosphorus adsorption on 2MgO/BC-450-0.5 was proposed based on the above results. When 2MgO/BC-450-0.5 was loaded in phosphorus solution, MgO nanoparticles on the surface of 2MgO/BC-450-0.5 were rapidly protonated under acidic conditions (pH = 5.2), resulting in the formation of MgOH^+^, which further reacted with adsorbed phosphate ions to produce Mg(H_2_PO_4_)_2_ and MgHPO_4_ crystals [54]. The adsorption processes can be summarized as follows:
Protonation process:
Mg − O + H_2_O→Mg − OH^+^ + OH^−^Electrostatic attraction:
Mg − OH^+^ + H_2_PO_4_^−^→Mg − OH^+^—H_2_PO_4_^−^Mg − OH^+^ + HPO_4_^2−^→Mg − OH^+^—HPO_4_^2−^Precipitation process:
Mg^2+^ + H_2_PO_4_^−^→Mg(H_2_PO_4_)_2_Mg^2+^ + HPO_4_^2−^→MgHPO_4_


### 3.8. Recycling Stability of 2MgO/BC-450-0.5

The recycling adsorption experiment was conducted for 12 h with initial phosphorus concentration of 10 mg L^−1^ and 2MgO/BC-450-0.5 dosage of 2 g L^−1^. As shown in Figure 7, phosphorus removal efficiency was slightly reduced from 99.32% to 84.17% after 5 cycles, demonstrating that 2MgO/BC-450-0.5 had a good recycling stability. The decrease in adsorption capacity might be attributable to the residual phosphorus on the surface of 2MgO/BC-450-0.5 through strong bonds between phosphorus and the adsorbent [62]. It can be concluded that MgO particles can be uniformly and stably loaded on the biochar, suggesting that ball milling is a simple and potential approach to modify biochar with metal oxides.

### 3.9. Treatment of Livestock Wastewater with 2MgO/BC-450-0.5

In order to evaluate the performance in the treatment of actual wastewater, 0.25–0.75 g L^−1^ of 2MgO/BC-450-0.5 was loaded in the livestock wastewater containing 39.51 mg L^−1^ phosphorus. After treatment for 12 h, phosphorus removal efficiency of 63%, 85% and 100% were obtained with 0.25, 0.5 and 0.75 g L^−1^ of absorbent (Figure 8), indicating that 2MgO/BC-450-0.5 is a promising adsorbent for removing phosphorus from livestock wastewater to meet discharge standards.

## 4. Conclusions

The ball milling method has proved to be highly efficient for biochar modification to enhance phosphorus adsorption performance. Mg precursor-to-rice straw mass ratio, pyrolysis temperature and ball milling time had a significant impact on the adsorption performance of final adsorbents. Typically, 2MgO/BC-450-0.5 prepared in optimum conditions showed the greatest phosphorus adsorption capacity and had good recycling stability. The kinetic behavior and adsorption isotherm of phosphate on 2MgO/BC-450-0.5 was fitted well by pseudo-second-order and Langmuir models, respectively. Adsorption of phosphorus on 2MgO/BC-450-0.5 was spontaneous and endothermic, mainly involving the protonation process, electrostatic attraction and precipitation process. This study demonstrated that the ball milling method is a potential strategy to prepare metal oxides/biochar composites in wastewater treatment.

## Figures and Tables

**Figure 1 ijerph-19-07770-f001:**
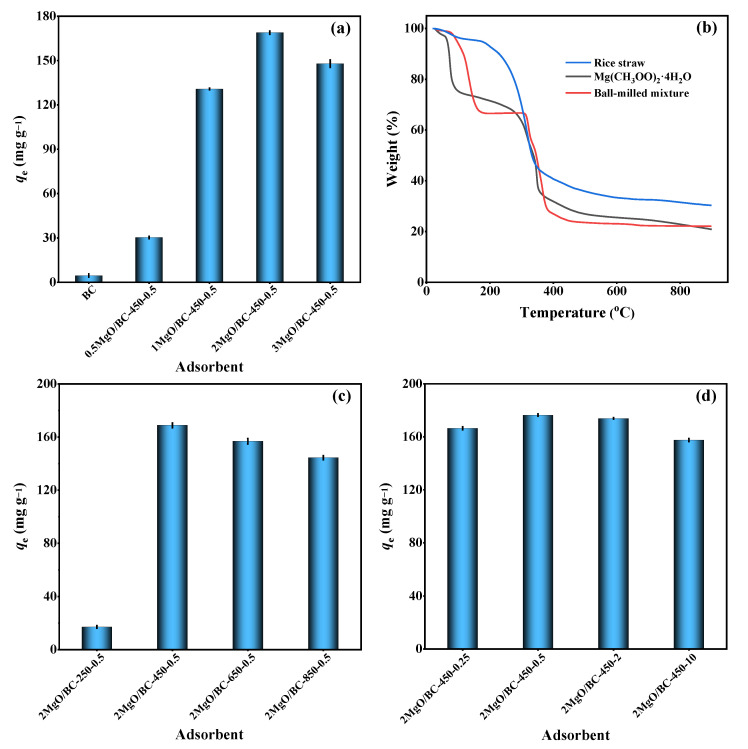
(**a**) Phosphorus adsorption on biochars prepared with different Mg precursor-to-rice straw mass ratios. (**b**) TGA curves of rice straw, Mg(CH_3_COO)_2_·4H_2_O and ball-milled mixture of Mg(CH_3_COO)_2_·4H_2_O and rice straw. (**c**,**d**) Phosphorus adsorption on MgO-modified biochars prepared at different pyrolysis temperatures and different ball milling times.

**Figure 2 ijerph-19-07770-f002:**
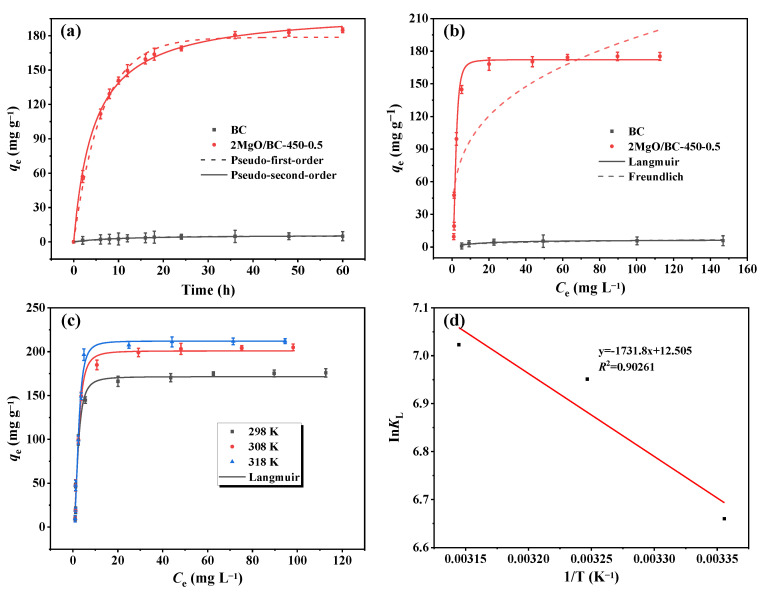
Adsorption kinetics (**a**) and isotherms (**b**) of phosphorus on BC and 2MgO/BC-450-0.5. Adsorption thermodynamics (**c**) and Va not Hoff plot (**d**) of phosphorus on 2MgO/BC-450-0.5.

**Figure 3 ijerph-19-07770-f003:**
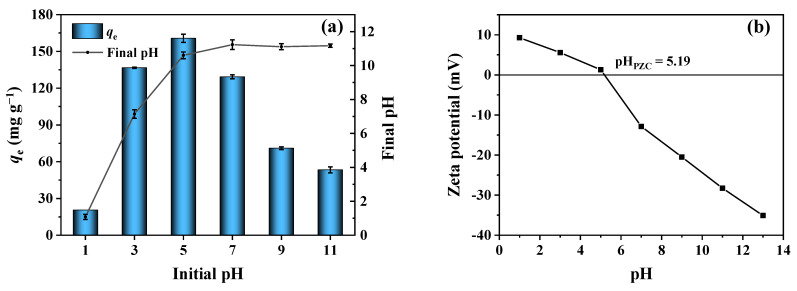
The effect of initial pH on phosphorus adsorption capacity (**a**) and Zeta potential determination for 2MgO/BC-450-0.5 (**b**).

**Figure 4 ijerph-19-07770-f004:**
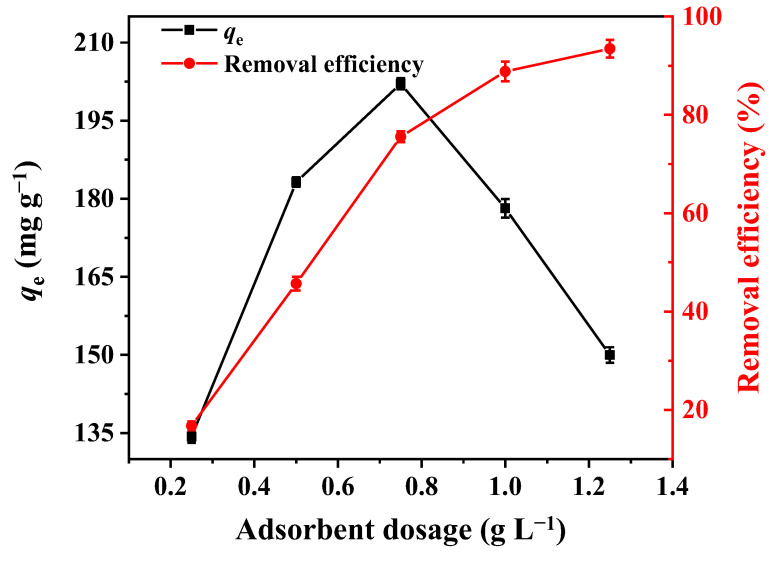
The effect of 2MgO/BC-450-0.5 dosage on phosphorus adsorption.

**Figure 5 ijerph-19-07770-f005:**
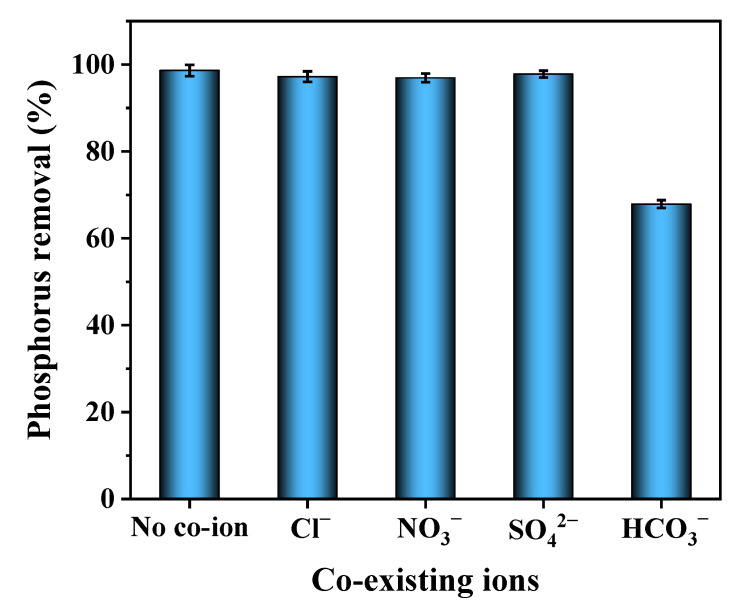
The effect of co-existing ions on phosphorus adsorption.

**Figure 6 ijerph-19-07770-f006:**
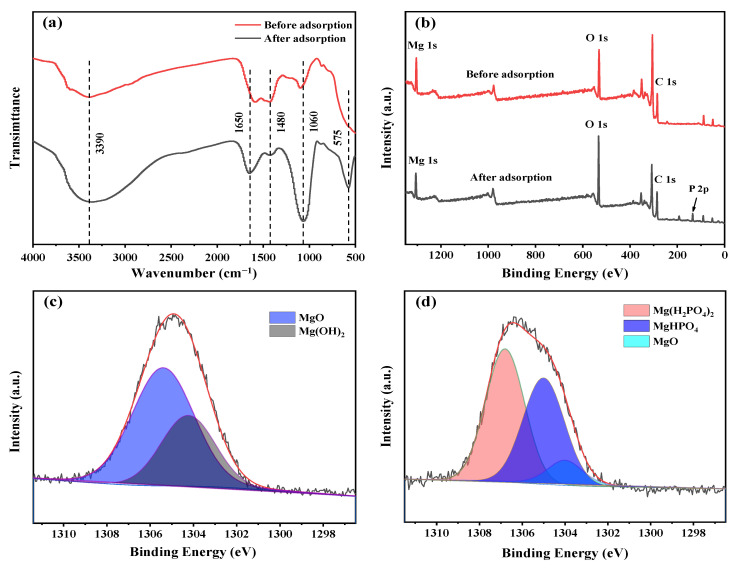
FTIR spectra (**a**) and XPS full-scan spectra (**b**) of 2MgO/BC-450-0.5 before and after adsorption. XPS spectra of Mg 1s before (**c**) and after (**d**) adsorption of 2MgO/BC-450-0.5.

**Figure 7 ijerph-19-07770-f007:**
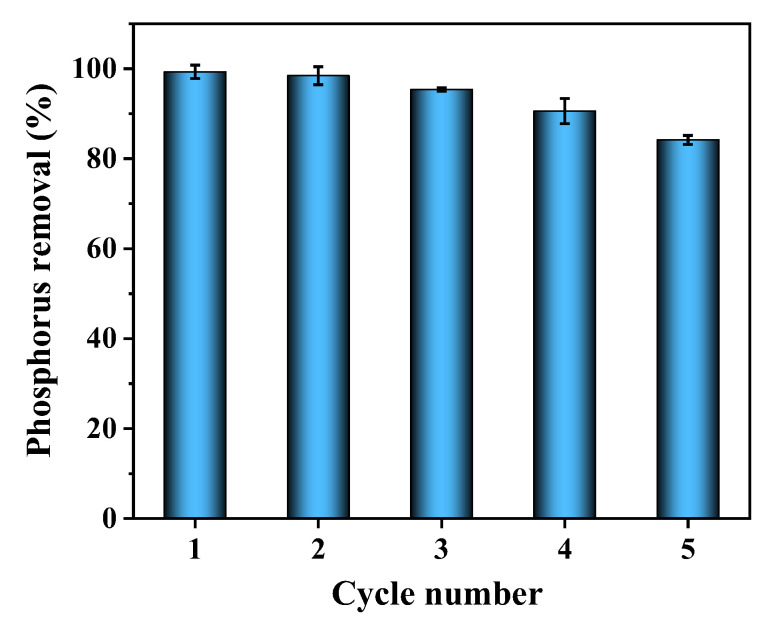
The recycling of 2MgO/BC-450-0.5 for phosphorus adsorption.

**Figure 8 ijerph-19-07770-f008:**
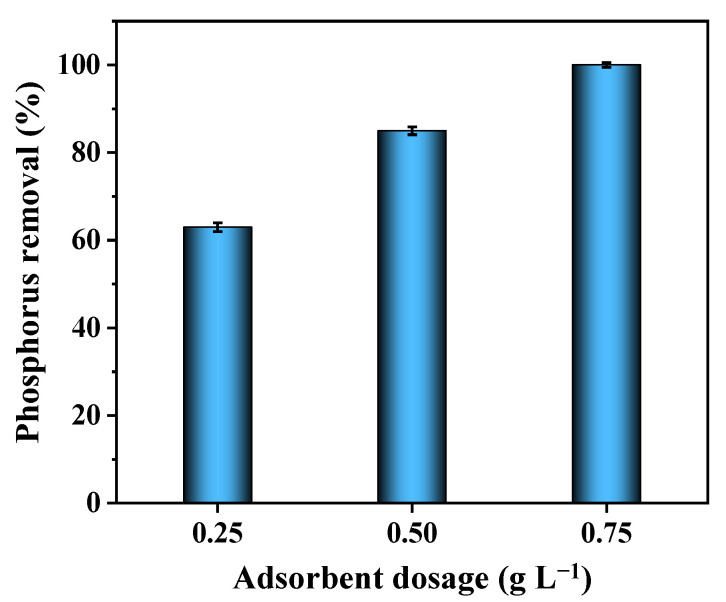
Adsorption of phosphorus from livestock wastewater with 2MgO/BC-450-0.5.

**Table 1 ijerph-19-07770-t001:** Physical properties of biochar samples.

Adsorbent	S_BET_ (m^2^ g^−1^)	S_micro_ (m^2^ g^−1^)	S_external_ (m^2^ g^−1^)	V_total_ (cm^3^ g^−1^)	D_BJH_ (nm)
BC	12.31	7.22	5.09	0.017	18.19
0.5MgO/BC-450-0.5	20.50	6.51	13.99	0.024	7.96
1MgO/BC-450-0.5	71.92	22.87	49.05	0.051	4.74
2MgO/BC-450-0.5	144.87	9.03	135.83	0.197	6.12
3MgO/BC-450-0.5	122.12	14.09	108.02	0.190	7.65
2MgO/BC-250-0.5	40.10	1.39	38.71	0.142	2.16
2MgO/BC-650-0.5	164.18	45.46	118.72	0.432	12.95
2MgO/BC-850-0.5	212.65	43.74	168.91	0.567	13.15
2MgO/BC-450-0.25	57.20	1.18	56.02	0.090	6.49
2MgO/BC-450-2	132.59	4.24	128.35	0.186	6.30
2MgO/BC-450-10	72.72	1.72	71.01	0.085	4.79

**Table 2 ijerph-19-07770-t002:** The maximum phosphorus adsorption capacity of different adsorbents at room temperature from existing literature and in this work.

Adsorbent	The Ratio of Solid to Solution	*q*_max_ (mg g^−1^)	Metal Oxide Content	References
2MgO/BC-450-0.5	20 mg:40 mL	171.54	MgO, 46.03%	This work
20MMSB	50 mg:20 mL	121.25	MgO, 34.12%	[22]
OMC-MgO-T800	20 mg:20 mL	107.00	MgO, 16.83%	[47]
4MCB	100 mg:30 mL	60.95	MgO, 28.53%	[50]
Ca-rich biochar	100 mg:50 mL	153.85	CaO, 25.98%	[51]
BC20	1000 mg:100 mL	147.06	CaO, 16.13%	[52]
LMZ	100 mg:40 mL	52.25	La_2_O_3_, 28.80%	[53]

## Data Availability

The data presented in this study are available on request from the corresponding author.

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
