# Peer review of "Solvent-Free Synthesis of MgO-Modified Biochars for Phosphorus Removal from Wastewater"

_ijerph, 2022, doi:10.3390/ijerph19137770_

Round 1

Reviewer 1 Report

This manuscript reported a solvent-free synthesis of MgO-modified biochar for phosphorus removal from wastewater. The strategy proposed by authors shows several merits compared to the existing strategies. The as-prepared biochar composites showed a high phosphorus adsorption performance, even in the livestock wastewater. In addition, various characterizations were conducted to show the structure and phosphorus adsorption mechanisms. However, minor revision is needed for this manuscript.

Specific comments:

1. Keywords: “metal oxides” or “magnesium oxide”?

2. Subsection 2.4.2: The introduction about the equations for kinetics, isotherms, and thermodynamics can be removed to Supporting Information. Because these are commonly used, not new equations.

3. Please use “–” to represent chemical bond, not “-”. For example, in line 327 “-OH” should be “–OH”.

4. In FTIR spectra, a band at around 3700 cm-1 can be assigned to free –OH on the surface of MgO, which was disappeared after adsorption. This change can be a hint for the involution of MgO in phosphate adsorption.

Reviewer 2 Report

This paper reports removal of phosphorus from waste water by MgO-modified biochars which were prepared by solvent-free ball milling method. Experiments are properly performed to lead to reasonable discussion and conclusion. I recommended to receive this paper after some minor revisions. Some issues needed to be answered were shown below:

1.      In the “Recycling stability of 2MgO/BC-450-0.5” section, phosphorus removal efficiency was slightly reduced from 99.32% to 84.17% after 5 cycles. What are the possible reasons?

2.      Line 191 should be revised. What does “mixing ball-milling of biochar and MgO” mean? Please check.

3.      More information about mechanochemical ball milling technology is suggested to be provided in the introduction.

4.      The Mg content in MgO/BC should be provided, at least for the sample 2MgO/BC-450-0.5.

5.      The adsorption time for livestock wastewater in experimental section is 24 h (line 173). However, the value is 12 h in discussion section (line 377). Please check.

6.      There are some typos. In line 16, “m3” should be modified to “cm3”. In line 22, “studied” should be “studies”.

Reviewer 3 Report

An environmentally relevant wastewater treatment method with adsorption is discussed. The manuscript focuses on analyzing a new preparation method and its effect on phosphorus removal. Several relevant parameters in the preparation and adsorption process were studied experimentally and analyzed both with adsorption models (isotherms and kinetics) as well as from chemistry viewpoints. In my opinion, the experimental part is here of higher quality than the model-based analysis. However, the manuscript deserves publication after the following concerns have been addressed:

1. Various forms in which phosphorus exists (phosphates etc.) were discussed later in the manuscript, but this could be mentioned already in the Introduction

2.  There are some weird expressions, although text in general is good. E. g. page  lines 56-57, “relatively excellent” and “Typically, Li et al. used…”. Later on the same page, “detailed studied”. On p. 4, line 168 “destroy the combination of phosphorus and…”; what kind of combination?

3.       On p. 3 the reference case of not milling is not indicated in the milling time list. Later in Fig 1 milling time is indicated in the figure caption, but there seems to be something wrong in the figure (no time mentioned).

4.       In the adsorption kinetic models, please show also original differential form indicating time dependency of adsorbed quantity. It seems that the integral forms are derived by using constant driving force corresponding to time t (adsorbed amount qt in the equation), which is not correct. It seems that logarithm is also indicated with in (with capital i), not ln. This chapter should also include proper literature references.

5.       Check usage of Langmuir parameter in thermodynamical analysis (see e.g. Journal of Molecular Liquids Volume 225, January 2017, Pages 137-146)

6.       Ch. 3.1. would benefit some sub-headings to organize the text according to the tested parameter.

7.       P. 5, some more information could be given about connection between TGA and pyrolysis. Were they carried out in similar conditions etc.?

8.       When various parameter effects are tested, it would be good to emphasize more which was the base case, i.e. what was kept constant.

9.       In the figures there seems to be tiny error bars. How were those determined? No information about repeated experiments was given. Are the error bars calculated with analysis uncertainty only? There are probably much wider variations originating from other sources related to raw material and preparation. This variation is important when conclusions are drawn whether changing some parameter has an effect or not.

10.   MgO vs phosphate stoichiometry could be briefly discussed.

11.   P. 7, line 270. Irreversibility as concluded from the thermodynamic parameters may be confusing as compared to cyclic operation tests.

12.   Fig 2: It is doubtful if the Langmuir isotherms are properly identified from the present data. There should be more experimental points around concentrations where the curve bends. This is essential also for the thermodynamical analysis and its error analysis.

13.   Conclusions in Ch 3.4 seem doubtful. Changing adsorbent to liquid ratio should lead to similar data as changing initial concentration, and it should be possible to identify isotherm parameters from this data also, provided that material balance for the liquid is properly expressed (equilibrium concentration in the liquid, not initial concentration). This is related to the previous comment.

14.   Ch. 3.7. Some discussion about the feasibility regarding number of recycles could be given. How many times the adsorbent should be recycled in practical applications?

15.   Ch. 3.8. How would adsorption isotherm and material balance predict removal efficiency in real wastewaters?

Round 2

Reviewer 3 Report

My comments are satisfactorily answered, and in my opinion the manuscript could be published. The main strengt seems to remain in the experimental part, while modeling mainly supports some experimental conclusions.